# Beat frequency quartz-enhanced photoacoustic spectroscopy for fast and calibration-free continuous trace-gas monitoring

Hongpeng Wu[1,2,3,*], Lei Dong[1,2,*], Huadan Zheng[1,2,3], Yajun Yu[3], Weiguang Ma[1,2], Lei Zhang[1,2], Wangbao Yin[1,2], Liantuan Xiao[1,2], Suotang Jia[1,2] & Frank K. Tittel[3]

Quartz-enhanced photoacoustic spectroscopy (QEPAS) is a sensitive gas detection technique which requires frequent calibration and has a long response time. Here we report beat frequency (BF) QEPAS that can be used for ultra-sensitive calibration-free trace-gas detection and fast spectral scan applications. The resonance frequency and $Q$-factor of the quartz tuning fork (QTF) as well as the trace-gas concentration can be obtained simultaneously by detecting the beat frequency signal generated when the transient response signal of the QTF is demodulated at its non-resonance frequency. Hence, BF-QEPAS avoids a calibration process and permits continuous monitoring of a targeted trace gas. Three semiconductor lasers were selected as the excitation source to verify the performance of the BF-QEPAS technique. The BF-QEPAS method is capable of measuring lower trace-gas concentration levels with shorter averaging times as compared to conventional PAS and QEPAS techniques and determines the electrical QTF parameters precisely.

[1] State Key Laboratory of Quantum Optics and Quantum Optics Devices, Institute of Laser Spectroscopy, Shanxi University, Taiyuan 030006, China. [2] Collaborative Innovation Center of Extreme Optics, Shanxi University, Taiyuan 030006, China. [3] Department of Electrical and Computer Engineering, Rice University, 6100 Main Street, Houston, Texas 77005, USA. * These authors contributed equally to this work. Correspondence and requests for materials should be addressed to L.D. (email: donglei@sxu.edu.cn).

Chemical gas phase analysis is significant in physics, chemistry, atmospheric science, space science, bioengineering, the life sciences as well as in medical applications[1–4]. Fundamental requirements for chemical sensing are sensitivity, selectivity and affordability. Quartz-enhanced photoacoustic spectroscopy (QEPAS), a variant of PAS, is useful as it meets these criterias[5]. Compared to PAS, the key innovation of QEPAS is to convert the acoustic wave induced by a modulated laser beam absorbed by the analyte into an electrical signal by means of a low-cost quartz tuning fork (QTF) based on its piezoelectric properties[6]. Such QTFs are mass-produced as a frequency reference and widely employed in cell phones as well as watches. QEPAS-based sensors combine the two main characteristics of PAS, that is, excitation wavelength independence and a detection sensitivity, which is proportional to the incident laser power[7]. QEPAS benefits from a set of unique piezoelectric properties of a QTF, such as a small size and a narrow acoustic resonance, resulting in an extremely high quality factor (10,000–15,000 at atmospheric pressure) as well as immunity to environmental acoustic noise[6].

QEPAS-based sensors have been demonstrated for the detection of numerous inorganic and organic trace gases using a variety of laser sources in the wavelength range that covers the ultraviolet[8], visible[9], near-infrared[10], mid-infrared[11] and the terahertz spectral regions[12,13]. In recent years, several variants of QEPAS have been developed, including SO-QEPAS[14], double acoustic micro-resonator (AmR) QEPAS[15], intra-cavity QEPAS[16,17] and evanescent-wave QEPAS[18]. The performance of different QEPAS-based sensor systems can be evaluated and compared based on the normalized noise equivalent absorption coefficient (NNEA) with respect to the optical laser power, the absorption line intensity and the detection bandwidth[19]. To date the lowest NNEA coefficient of QEPAS sensors, $\sim 10^{-10}\,\mathrm{cm}^{-1}\,\mathrm{W}\,\mathrm{Hz}^{-1/2}$ was obtained for $SF_6$ detection, which is one order of magnitude lower than the best NNEA values reported for conventional PAS results[20,21].

Requirements for on-line monitoring of trace gases are uninterrupted operation, fast response, high selectivity and high detection sensitivity as well as compact size with a small gas cell. In the majority of conventional QEPAS sensor systems reported to date, a commercially available 33 kHz QTF resonance frequency is employed. Two important parameters of the QTF must be determined repetitively via an electric excitation method[8], which defines the QEPAS sensor system calibration process. The first parameter is the resonance frequency $f_0$ of the QTF. The QTF has a frequency resonance width of 3–5 Hz at atmospheric pressure. Only frequency components in this resonance width can excite the vibrations of the QTF prongs efficiently[6]. The modulation of the laser radiation frequency ($f$) must accurately match with the QTF resonance frequency to obtain the highest signal amplitude. The second QTF parameter that must be measured is the $Q$-factor, as the magnitude of the $Q$-factor is proportional to the QTF signal amplitude[22].

Both $f_0$ and the $Q$-factor of a QTF are subject to the fabrication process[23] and also depend on the operating environment, which includes the gas pressure, the temperature and the gas composition[24]. Thus in any trace-gas monitoring application, these two QTF parameters have to be determined prior to a QEPAS measurement and verified after a sensor operated for more than 24 h or if there is an environmental temperature change of $>3\,°C$ (ref. 25). Such a calibration process, which usually requires about 90 s to complete, interrupts a continuous QEPAS measurement. Furthermore, an inherently high $Q$-factor of the QTF provides the high sensitivity of conventional QEPAS, but also leads to a long accumulation time of the acoustic energy[26]. According to classical oscillator theory, the response time $\tau$ is given by

$$\tau = \frac{Q}{\pi f_0} \qquad (1)$$

The QTF response time is typically about 100 ms and the measurement period can be up to 300–400 ms for uncorrelated trace-gas concentration measurements. This time interval is too long to perform rapid trace-gas measurements which are important in a wide range of industrial applications[26], especially for real-time monitoring of trace gases.

In this article, we report an innovative spectroscopic measurement technique that relies on the beat frequency (BF) signal between the QTF resonance frequency $f_0$ and the laser modulation frequency $f$. The basic concept of BF-QEPAS requires that the laser modulation frequency has a frequency difference of $\Delta f$ with respect to the QTF resonance frequency $f_0$. For this condition, a BF-QEPAS signal with a period of $\Delta f$ is generated and observed when the wavelength of the excitation laser is rapidly scanned across a targeted absorption line via a pulsed ramp current and the QTF signal is demodulated at the laser modulation frequency $f$. This technique can acquire the resonance frequency $f_0$ and $Q$-factor of the QTF as well as the trace-gas concentration information in a measurement time of about 30 ms.

## Results

**Theory of BF-QEPAS**. The QTF can be modelled as a series resistance–inductor–capacitor circuit (based on the RLC Butterworth–Van Dyke model) using an electrical analogy of a mechanical damped oscillator[27–29]. This can be expressed by Newton's law of motion:

$$L\frac{d^2q}{dt^2} + R\frac{dq}{dt} + q\frac{1}{C} = U(t) \qquad (2)$$

in which the resistance ($R$) represents the acoustic losses in the material and its environment, the inductor ($L$) represents the mass of the oscillator, the inverse capacity ($1/C$) and the electrical charge ($q$) represent the stiffness and displacement of the mechanical oscillator. The voltage ($U$) represents the force applied to the mechanical oscillator. The solution of equation (2) has the following form:

$$q(t) = q_s(t) + q_t(t) \qquad (3)$$

The term $q_s(t)$ accounts for the steady state response and $q_t(t)$ provides the transient response of the system. Figure 1 depicts a side-by-side principle comparison for conventional QEPAS and BF-QEPAS techniques using the steady and transient response of the QTF, respectively.

In conventional QEPAS, a slowly varying continuous acoustic wave causes forced vibrations of the QTF. The transient response of the QTF is neglected and only the steady state behaviour is taken into account due to the long averaging time ($>300$ ms) used during which the transient response is averaged to be zero. In BF-QEPAS, an acoustic pulse induced by the target gas absorption is generated as a result of rapid wavelength scanning ($>30\,\mathrm{cm}^{-1}\,\mathrm{s}^{-1}$), which causes the prongs of the QTF to vibrate in a short period of time. Subsequently the QTF prong changes to a free vibration mode after the acoustic pulse terminates rather than the continuous forced vibrations caused by a continuous acoustic wave as in conventional QEPAS. The vibration energy will be dissipated via extrinsic and intrinsic QTF loss mechanisms[30]. At this point, the QTF is vibrating at its resonance frequency and not at the laser modulation frequency. The QTF signal is demodulated at the laser modulation frequency $f$. When the averaging time is short enough ($<100$ ms) to provide a sufficient system detection bandwidth, a BF signal with an

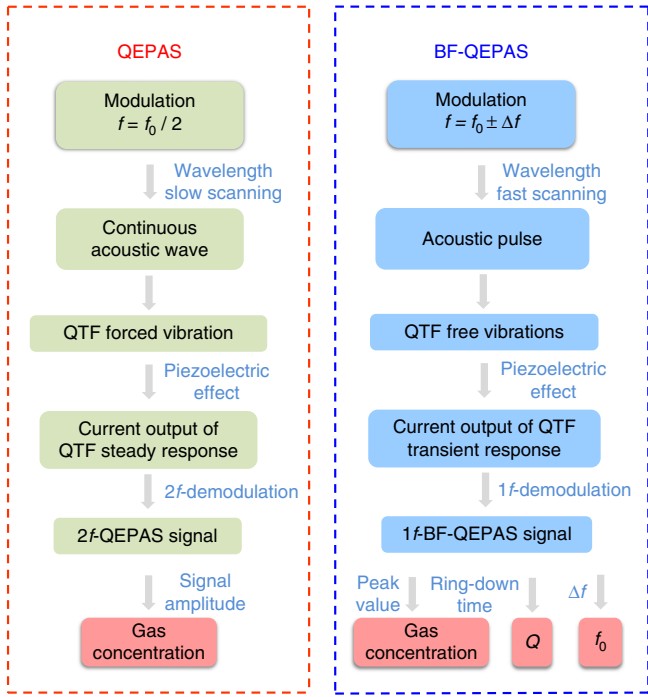

**Figure 1 | A side-by-side comparison of conventional QEPAS and BF-QEPAS techniques.** Unlike conventional QEPAS, the modulation frequency $f$ of the laser in the BF-QEPAS technique is shifted from the QTF resonance frequency $f_0$. The laser wavelength is rapidly scanned with respect to the QTF response time. By analyzing the beat signal generated between the laser modulation frequency and the QTF resonance frequency, a BF-QEPAS-based sensor can determine the target gas concentration, the QTF resonance frequency and the $Q$-factor in a single measurement.

exponential decay envelope is generated from the QTF-transient response. The concentration of the trace gas, the resonance frequency and $Q$-factor of the QTF can be obtained by measuring the peak amplitude, the BF $\Delta f$ and the decay time $\tau$ of the BF-QEPAS signal, respectively.

**Experimental apparatus.** A schematic of the experimental set-up for demonstrating the performance of the BF-QEPAS spectrophone is shown in Fig. 2. A 1,368.7 nm distributed feedback diode laser (DFB-DL; NTT Electronics, Inc. Model NLK1E5E1AA) was employed as the excitation source. The laser wavelength can be tuned by varying its current and temperature. The laser temperature was stabilized by a temperature controller (THORLABS, Inc. Model TED 200C). The laser drive current consisted of three components: a direct current (d.c.), an alternating current (a.c.) and a ramp signal. The d.c. component from a current source (ILX Lightwave Corp. LDX 3220) determined the centre wavelength of the DFB-DL. The continuous sinusoidal a.c. component, generated from the function generator 1 (Stanford Model DS345) realized the wavelength modulation while a ramp signal, provided by a second function generator 2 (Tektronix, Inc. Model AFG 3022), scanned across the targeted trace-gas absorption line and then remained at a constant value to complete the induced BF-QEPAS signal detection. The scanning cycle is the sum of the scanning time and the waiting time. The repetition rate of the ramp signal is 12 Hz. The waiting time of a single scanning cycle is 0.05 s. A fibre-coupled collimator (OZ optics Ltd. Model LPC-01) produced a diode laser beam with a 200 μm diameter, which was directed to an acoustic detection module (ADM) and avoided touching the QTF[31]. For comparison, the ADM with an on beam configuration (Methods section), which is widely used in

conventional QEPAS sensors, is placed in the BF-QEPAS experimental set-up. Although there is no standing acoustic wave built up in the acoustic resonator with the excitation of an acoustic pulse, the behaviour of signal enhancement is expected as the non-resonant micro-tube can effectively confine the acoustic pulse[21]. The ADM is enclosed in a gas enclosure with two CaF₂ windows. A vacuum pump and a pressure controller (MKS Instruments Inc. Model 649B13TS1M22M) controlled and maintained the pressure inside the ADM. The gas flow was set to 120 s.c.c.m. by a pin valve to minimize flow noise. The piezoelectric signal generated by the QTF was processed by a transimpedance amplifier with a 10 MΩ feedback resistor $R_g$ and directed to a lock-in amplifier (LIA; Stanford Research Systems, Model SR830). The harmonic signal demodulated by the LIA was then transmitted to computer via a DAQ card (National Instruments, Inc., Model BNC-2110).

**BF-QEPAS signal simulation and analysis.** The performance of the BF-QEPAS technique was simulated using MATLAB software (Supplementary Note 1). In Fig. 3a, a QTF resonance frequency, $f_0 = 32,760$ Hz, was employed as the laser modulation frequency and the interaction time $t_a$ between the acoustic wave and the QTF was long enough ($t_a = 10$ s) to provide a continuous acoustic wave with a slow amplitude variation. The integration time was set to 300 ms corresponding to a detection bandwidth of 0.417 Hz. An excellent 1$f$-based conventional QEPAS signal can be obtained as shown in Fig. 3a. In Fig. 3b, the value of $t_a$ was decreased to 500 ms and the laser modulation frequency was set to the QTF resonance frequency, $f_0$. Two decaying tails, related to the peak and the valley of the first harmonic signal were observed when the LIA detection bandwidth was 0.417 Hz as previously, which implies that the QTF-transient response gradually dominates in the excitation of a quasi-pulsed acoustic signal of the QTF. But the first decaying tail was influenced by the valley of the first harmonic signal and only the second free-decaying tail caused by the valley reflects a correct response time. When the interaction time was decreased to 10 ms, an acoustic pulse was produced due to an interaction time that is much shorter than the QTF response time. When the modulation frequency ($f = 32,960$ Hz) was detuned from the QTF resonance frequency, the BF-QEPAS signal as shown in Fig. 3c was observed. In this case, the averaging time was reduced to 100 μs (a detection bandwidth of 1,250 Hz) to provide a sufficient response bandwidth and to maintain efficient background noise suppression. In addition, the result showed that the amplitude of the BF-QEPAS signal is proportional to the strength of the acoustic force, which is determined by the target gas concentration. This gas concentration can be obtained by detecting the peak value of the BF-QEPAS signal. The value of $f_0$ can be obtained since the BF is equal to the difference between $f_0$ and $f$ values:

$$\Delta f = \frac{1}{\Delta t} = |f - f_0| \qquad (4)$$

where $\Delta t$ is the time interval between two adjacent peaks or valleys of the BF signal. As the $Q$-factor can be defined as the ratio of the energy accumulated in a resonator to the energy loss per cycle, the $Q$ can be calculated using equation (1), from which $\tau$ is obtained by an exponential decay fitting of the BF-QEPAS envelope as in ring-down spectroscopy[32]. The simulated results, $f_s = 32,762$ Hz and $Q_s = 1,882$, are consistent with the experimental results, $f_0 = 32,760$ Hz and $Q = 1,846$, obtained by the electric excitation method[8].

The experiment based on the BF-QEPAS technique was carried out using the corresponding detection bandwidths used in simulations and the apparatus depicted in Fig. 2. The experimental results, shown in Fig. 3d–f, are in excellent agreement

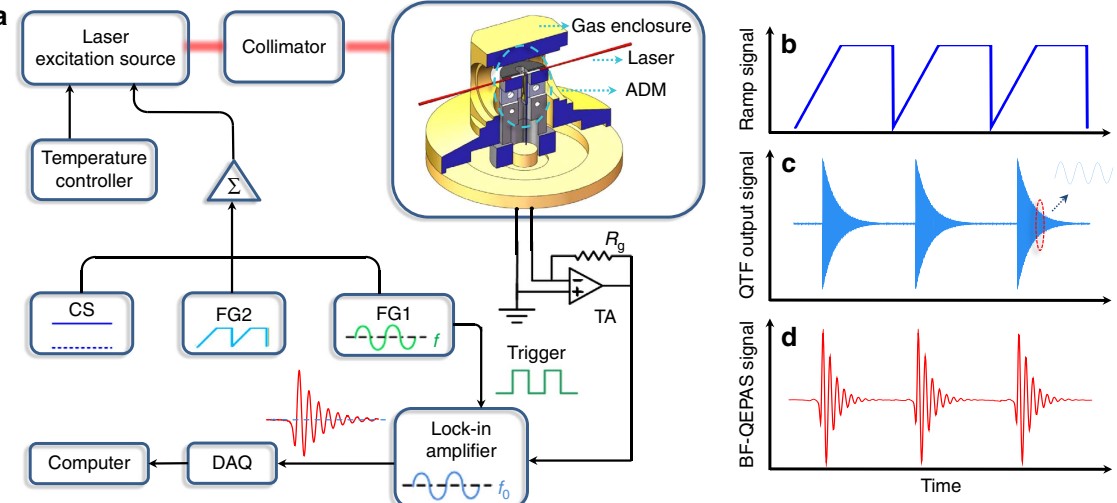

**Figure 2 | Schematic of the experimental BF-QEPAS apparatus. (a)** The diode laser was operated by means of a current and temperature controller. A direct current (d.c.), alternating current (a.c.) and ramp signal provided by current source, function generator 1 (FG1) and function generator 2 (FG2), respectively, were used as the laser drive current, modulation current and scanning current, respectively. Three different semiconductor lasers, DFB-DL, DFB quantum cascade laser (DFB-QCL) and DFB interband cascade laser (DFB-ICL), were employed in this system as the excitation sources sequentially. A fibre-coupled collimator ensures that the collimated DFB-DL beam passes through the ADM without touching the QTF prongs. Optical lenses were used to collimate the DFB-ICL and DFB-QCL laser beams. The details about the experiments, in which the DFB-QCL and DFB-ICL were equipped as the excitation source, were described in the Supplementary Figs 6 and 7, respectively. DAQ, data acquisition; TA, transimpedance amplifier. **(b)** The ramp signal provided by FG2. **(c)** The output signal generated by the piezoelectric effect of the QTF after its prongs were excited by an acoustic pulse. **(d)** The BF signal generated after the piezoelectric signal was demodulated by a LIA.

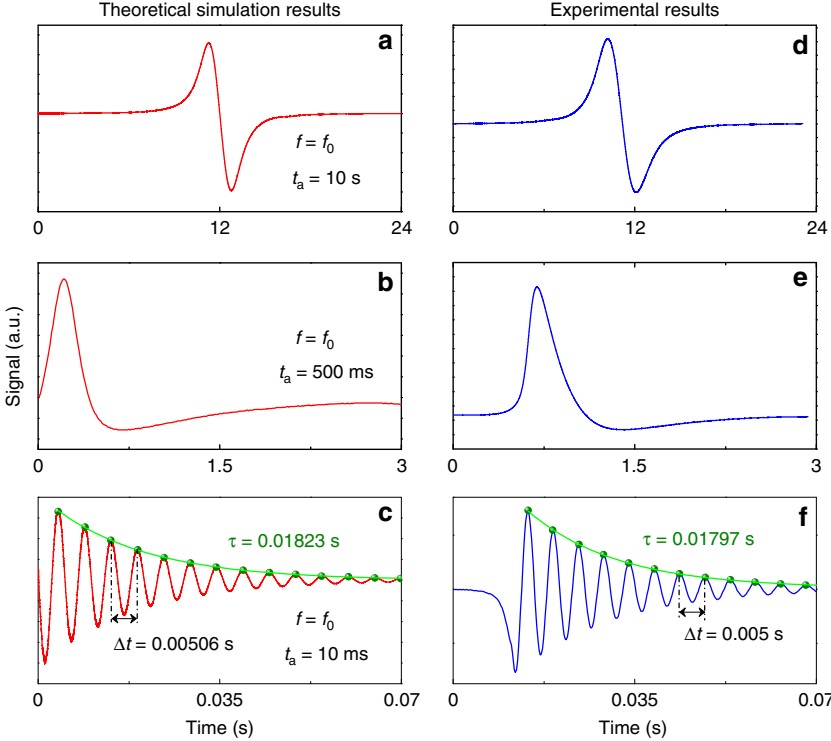

**Figure 3 | Simulation and experimental results of BF-QEPAS. (a–c)** First harmonic QTF output signal for different modulation frequencies and wavelength-scanning rates were simulated by MATLAB software with actual parameters of the QTF system. The different wavelength-scanning rates can be simulated by changing the value of $t_a$ as this parameter represents the action time of the acoustic force to the QTF. The value of $t_a$ was estimated by using the ratio of the absorption line width to the wavelength scanning rate. **(d–f)** The corresponding tests were carried out with 2.5% water vapour at room temperature and atmosphere pressure. The wavelength was scanned at a rate of $0.12\,cm^{-1}s^{-1}$, $3\,cm^{-1}s^{-1}$ and $72\,cm^{-1}s^{-1}$, respectively by scanning the laser current. **(a,b,d,e)** The modulation frequency of the laser current was 32,760 Hz, while for **c,f** it was 32,960 Hz.

with the simulated results in Fig. 3a–c. As the DFB-DL wavelength scan rate varied from 0.12 cm$^{-1}$s$^{-1}$ to 3 cm$^{-1}$s$^{-1}$, the conventional QEPAS signal changed from a standard first harmonic signal to an irregular signal with an exponentially decaying tail. A 200 Hz beat signal with a response time of 0.01797 s was observed, when the wavelength was scanned at the rate of 72 cm$^{-1}$s$^{-1}$ (equivalent to an interaction time of about 10 ms) and modulated at a frequency of 32,960 Hz,. The experimental results of the QTF parameter, $f_t = 32,760$ Hz and $Q_t = 1,848$ are in excellent agreement with the simulated results.

**Experimental optimization**. The performance of the BF-QEPAS sensor was optimized using 2.5% water vapour in N$_2$ (Methods section) at room temperature and atmosphere pressure. The selected H$_2$O absorption line is located at 7,306.75 cm$^{-1}$ with a line intensity of $1.8 \times 10^{-20}$ cm mol$^{-1}$. The laser temperature was set to 9.65 °C and the d.c. current was set to 120 mA to reach the H$_2$O target wavelength. The diode laser output power was 13.0 mW for these operating conditions. The filter slope and time constant of the LIA were set to 12 dB (Supplementary Fig. 2, Supplementary Note 2) and 100 µs (Supplementary Fig. 3 and Supplementary Note 2), corresponding to a detection bandwidth of 1,250 Hz. The experimental results verified that these settings do not distort the line shape of the BF-QEPAS signal and maintain the narrow bandwidth required for noise reduction.

In a conventional QEPAS-based sensor, the largest amplitude of the 2f harmonic component is generally lower than the 1f signal amplitude, as shown in Fig. 4a,b. However, the 2f harmonic component is often used as the detection signal. This is due to the fact that the peak position of the 2f signal corresponds to that of the target absorption line[33]. Furthermore, the long response time τ of the QTF requires that the conventional QEPAS-based sensors must be operated in a line-locking mode[21]. In the novel BF-QEPAS-based sensor, the BF-QEPAS signal from the first, second, third harmonic have the same wave shape. The optimum harmonic must be determined experimentally. The BF-QEPAS signals as shown in Fig. 4d–f from the first, second, third

harmonics at a detection bandwidth of 1,250 Hz were measured. Further evaluation tests of the BF-QEPAS were performed at the first harmonic, since the 1f BF-QEPAS signal has the largest amplitude and obtains an improved minimum detection limit.

The modulation frequency and depth have a significant impact on BF-QEPAS signals. Optimization of the modulation frequency was carried out with the same modulation depth (MD = 29 mA) and an optimal wavelength scan rate, 36 cm$^{-1}$s$^{-1}$ (Supplementary Fig. 4 and Supplementary Note 2). The optimization results are plotted in Fig. 5a. The measurement was implemented from 32,440 Hz to 33,075 Hz as the QTF does not respond to frequencies outside of this range. The BF-QEPAS signal amplitudes between 32,440 and 32,705 Hz were analyzed, since the response curve of the BF-QEPAS signal amplitude is symmetric and centred at the QTF resonance frequency. Two factors determine the amplitude of the BF-QEPAS signals. The first factor is the response of the QTF to the acoustic wave frequency which is equal to the laser modulation frequency. The QTF is easier excited to vibrate, when the laser modulation frequency $f$ approaches the QTF resonance frequency $f_0$. The second factor is the value of the BF, Δ$f$. A small BF causes a large interval between two constructive interferences. The longer the interval is, the more vibrating energy of the QTF is dissipated during the interval. These two factors work in unison to generate a maximum amplitude of the BF signal. In Fig. 5a, the amplitude of the BF-QEPAS signal increases in the frequency range from 32,440 to 32,640 Hz as the response of the QTF is enhanced, while the amplitude decreases between 32,640 and 32,705 Hz, because the second factor gradually dominates. In addition, the MD was optimized at the optimal modulation frequency, $f = 32,640$ Hz. A maximum BF-QEPAS signal was observed with a MD = 29 mA as shown in Fig. 5b. Hence, the following evaluation tests of this sensor were performed with a MD = 29 mA and $f = 32,640$ Hz.

For a comparison, the amplitude of the conventional 2f QEPAS signal as a function of frequency obtained with the same apparatus (Methods section) is also plotted in Fig. 5a. The peak of the 2f QEPAS signal at the resonance frequency is about 16% higher than the optimum amplitude of the BF-QEPAS

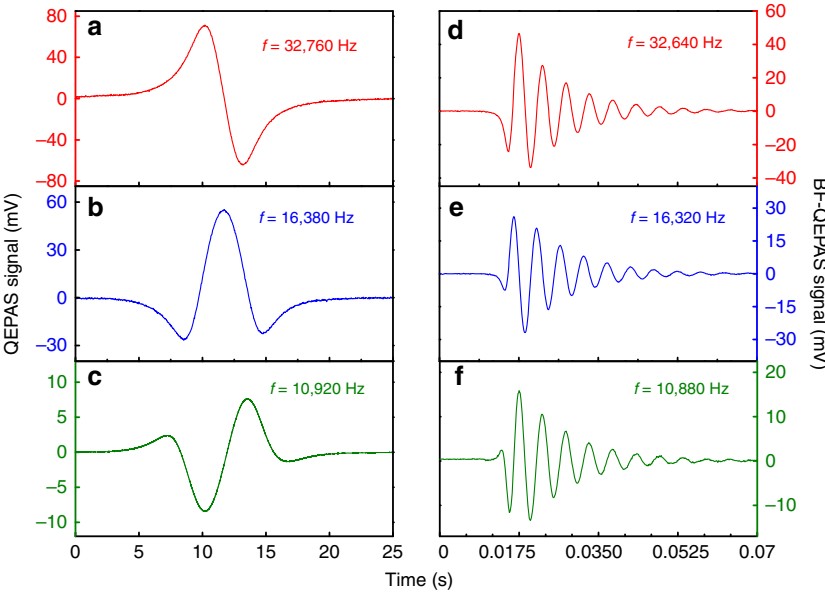

**Figure 4 | Standard first three harmonics and corresponding BF signals obtained with the conventional QEPAS and BF-QEPAS techniques.**
(**a–c**) The wavelength-scanning rate of the diode laser, the time constant and filter slope of the LIA for conventional QEPAS technique were 0.12 cm$^{-1}$s$^{-1}$, 300 ms and 12 dB (corresponding to a detection bandwidth of 0.417 Hz). (**d–f**) The same parameters were 36 cm$^{-1}$s$^{-1}$, 100 µs and 12 dB (corresponding to a detection bandwidth of 1,250 Hz) for the BF-QEPAS technique. The laser modulation frequencies for standard first (**a**), second (**b**) and third (**c**) harmonic were 32,760, 16,380 and 10,920 Hz, while they were 32,640, 16,320 and 10,880 Hz for the corresponding BF signals (**d–f**), respectively.

signal. However, the averaging time of the 1f BF-QEPAS is three orders of magnitude shorter than that for 2f QEPAS.

**Detection limit.** The detection sensitivity of a trace-gas sensor is determined by the signal-to-noise-ratio. Previous experimental studies verified that the background noise of a conventional QEPAS-based sensor, $\sqrt{\langle V_{N-R}^2 \rangle}$, is dominated by thermal noise of the QTF[34], which can be expressed as:

$$\sqrt{\langle V_{N-R}^2 \rangle} = R_g \sqrt{\frac{4k_B T}{R}} \sqrt{\Delta f_{det}} \quad (5)$$

where $k_B$ is the Boltzmann constant, $T$ is QTF temperature and $\Delta f_{det}$ is the total detection bandwidth of the system for the two LIA detection channels. For a BF-QEPAS-based sensor, the thermal noise of the QTF cannot be calculated using equation (5). Instead, the noise power density should be integrated over the QTF resonant curve, since the LIA bandwidth is significantly wider than the resonant curve of the QTF. The equivalent noise detection bandwidth (ENBW) of the QTF is $\pi f_0/2Q$. Hence the thermal noise of the QTF is expressed as

$$\sqrt{\langle V_{N-R}^2 \rangle} = R_g \sqrt{\frac{2\pi k_B T f_0}{RQ}} \quad (6)$$

The total fundamental noise also includes the feedback resistor noise $\sqrt{\left\langle V_{N-R_g}^2 \right\rangle}$ and should be integrated over the full lock-in detector bandwidth:

$$\sqrt{\left\langle V_{N-R_g}^2 \right\rangle} = \sqrt{4k_B T R_g \Delta f_{det}} \quad (7)$$

Hence the total noise $\sqrt{\langle V_N^2 \rangle}$ can be expressed as:

$$\sqrt{\langle V_N^2 \rangle} = \frac{1}{\sqrt{2}} \left( \sqrt{\langle V_{N-R}^2 \rangle} + \sqrt{\left\langle V_{N-R_g}^2 \right\rangle} \right)$$

$$= \frac{1}{\sqrt{2}} \left( R_g \sqrt{\frac{2\pi k_B T f_0}{RQ}} + \sqrt{4k_B T R_g \Delta f_{det}} \right) \quad (8)$$

The $1/\sqrt{2}$ coefficient reflects the fact that the noise is calculated only for one detection channel. The theoretically estimated noise level is 13.9 μV. The experimentally measured noise level was in the 11–15 μV range, which is in excellent agreement with the theoretically estimated result.

The linear response of the BF-QEPAS sensor was investigated by measuring the second peak amplitude of the 1f-based BF-QEPAS (Supplementary Fig. 5 and Supplementary Note 3) for different $H_2O$ concentrations (Methods section) with the optimized parameters reported above. The results were plotted in Fig. 6. The linearity of the sensor response to the $H_2O$ concentration levels was confirmed by the high $R^2$ value (>0.999) of the linear fitting.

The noise-equivalent concentration (NEC) of the BF-QEPAS-based sensor for 0.1 ms integration time was estimated to be about 5.9 p.p.m. based on the background noise level and the data depicted in Fig. 6. Hence, the NNEA coefficient for $H_2O$ is $2.45 \times 10^{-9}$ cm$^{-1}$ W Hz$^{-1/2}$ (Methods section), which is one order of magnitude higher than that obtained by the conventional QEPAS technique.

An Allan–Werle deviation analysis was performed in order to evaluate the long term stability of the BF-QEPAS sensor, as shown in Fig. 7. The ADM was filled with pure $N_2$ at atmospheric pressure. The sensor system was operated at room temperature in the BF-QEPAS mode and the laser wavelength was scanned at the optimized rate. The Allan–Werle deviation indicates that the thermal noise of the feedback resistor ($R_g$) and the QTF are the dominant noise sources and the BF-QEPAS-based sensor allows data averaging without a base line or sensitivity drift for a time scale of >70 s.

## Discussion

Three semiconductor laser sources were chosen as the excitation source with the same ADM to verify the performance of the BF-QEPAS technique. The laser beam from the three different excitation sources was focused by suitable lenses that ensure that the beam passes through the AmR tubes and QTF without

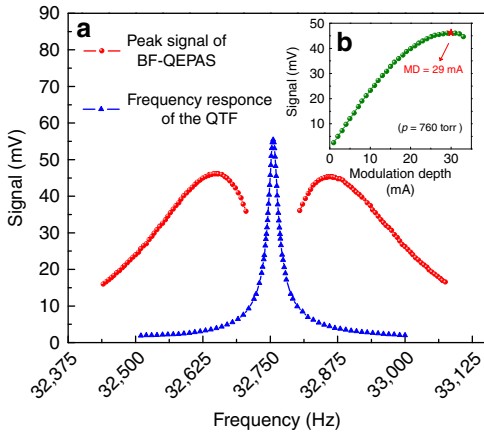

**Figure 5 | Qualitative representation of the QTF response curves. (a)** Red curve represents fits to the experimental data acquired via the BF-QEPAS technique, while the blue lines are fits to the peak value of the 2f signal generated by the conventional QEPAS technique. Both conventional QEPAS and BF-QEPAS signals are symmetrical and centred on the resonance frequency of the QTF. However, their maximum positions differ. The conventional QEPAS signal shows a Lorentzian-like behaviour. As a result, its maximum position is located at the resonance frequency of the QTF. The BF-QEPAS signal curve presents a two winged shape so that two maximum positions appear on both sides of the resonance frequency. **(b)** The amplitude of the beat signal as a function of the modulation depth (MD) current. The variation trend of the beat signal with the modulation depth increasing shows a similar behaviour as the conventional wavelength modulation technique. The signal amplitude rises when the modulation depth current is <29 mA. After reaching the maximum, the signal amplitude starts to decrease.

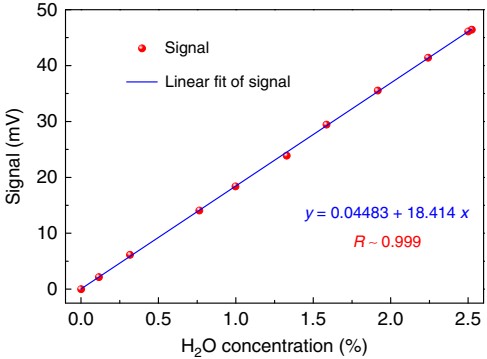

**Figure 6 | Linear dependence of the BF-QEPAS signal on $H_2O$ concentration levels.** The 1f-based BF-QEPAS signal was recorded as the $H_2O$ concentration levels were varied. For each concentration step, 50 readings of the BF-QEPAS signal were averaged to increase the accuracy of the result. The data was plotted as a function of $H_2O$ concentration, which confirms the linearity of the BF-QEPAS response to concentration.

touching. The experimental parameters, such as pressure, scan rate and modulation depth of the laser current were also optimized (Supplementary Figs 6 and 7 and Supplementary Note 4).

A side-by-side comparison of detection limits for the BF-QEPAS-based sensors and QEPAS-based sensors is shown in Table 1. The NEC was normalized to a 1 s integration time to facilitate inter-comparison. With three different excitation sources, the experimental results of the frequency and Q-factor of the QTF are consistent with the value obtained by conventional electric excitation method. The BF-QEPAS technique has an obviously lower NEC and NNEA value than the conventional QEPAS technique for the fast relaxing molecule ($H_2O$), while for the slow relaxing molecules ($CH_4$ and CO), the NEC and NNEA for BF-QEPAS technique are somewhat better than the conventional QEPAS technique. The BF-QEPAS technique similar to the conventional 2f wavelength modulation-based QEPAS technique provides high selectivity together with background-free signals. Its selectivity is mainly determined by an initial rapid wavelength scanning crossing the target absorption line. The target absorption line produces an acoustic pulse to push the QTF prongs to vibrate, ensuring high selectivity. Subsequently the high Q-factor of the QTF vibrating freely at its resonant frequency effectively suppresses the background noise from the non-resonant acoustic pulse. Only the intensity information of the acoustic pulse remains in the BF-QEPAS signal. As a result, the BF-QEPAS technique is dominated by the QTF thermal noise using either 1f or 2f detection.

In summary, an innovative modification of quartz enhanced photoacoustic spectroscopy PAS, BF-QEPAS, was demonstrated for the first time, which can simultaneously measure trace-gas concentrations, the QTF resonance frequency and Q-factor, thereby avoiding the calibration process and ensuring continuous real-time trace-gas monitoring. The BF-QEPAS technique acquires spectral data by detecting the beat signal generated by the transient response of the QEPAS system at a much faster rate than those reported in previous publications. The results show that this new technique is capable of providing reduced data acquisition times and improved detection sensitivity than those reported for conventional QEPAS, especially for fast relaxing molecules. The BF-QEPAS technique can be applied to all QEPAS-based sensor systems without any hardware changes of a conventional QEPAS sensor system. Further improvement of the detection sensitivity can be achieved either by optimizing the geometrical parameters of the non-resonant micro-tube or by combining with the overtone resonance mode of a custom fabricated QTF.

## Methods

**Preparation of an on-beam configuration ADM.** The on-beam configuration ADM consists of a standard commercial QTF and a set of AmRs (Supplementary Fig. 1). Two identical metallic tubes of 4.0 mm in length and 0.8 mm inner diameter were placed close to, but not touching, the QTF, with 20 μm gaps on both sides of the QTF. The vertical distance between the centre of the tubes and the QTF opening was 0.7 mm. In this case, the Q-factor of the QTF decreased to 1,850 as a result of coupling between the QTF and the AmRs[21,35]. The QTF and the AmR were mounted on an aluminum plate.

**Preparation of water vapour with different concentrations.** The highest water vapour mixing ratio reported in this paper, 2.524%, was obtained by controlling pure $N_2$ passing through a humidifier (PermSelect, PDMSXA-2500) at a stable flow rate, constant temperature and pressure. The other water vapour mixing ratios were generated by mixing $H_2O$ with pure $N_2$ at different flow rate ratios. Two mass flow metres (Alicat Scientific, Inc. Model M-2SLPM-D/5M) were used to control the flow rate. The concentration of water vapour was determined by direct absorption spectroscopy using a multi-pass gas cell with an effective path-length of 57.6 m described in ref. 36.

**Calculation of NNEA.** The NNEA coefficient can be determined by the following equation:

$$\text{NNEA} = \frac{\alpha_{\min}P}{\sqrt{\text{ENBW}}} \qquad (7)$$

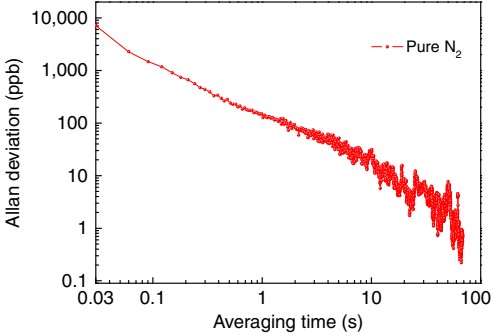

**Figure 7 | Allan–Werle deviation plot as a function of averaging time.** The background noise of the BF-QEPAS-based sensor was measured when the ADM was filled with pure $N_2$. The LIA time constant and filter slope as well as the wavelength-scanning rate of the diode laser were set to optimized parameters, 100 μs, 12 dB and 36 cm$^{-1}$ s$^{-1}$, respectively.

$\alpha_{\min}$ is the minimum detectable absorption coefficient, which can be calculated using the HITRAN database (http://www.hitran.com), if the absorption wavelength of the target trace gas and the NEC of the sensor are known. $P$ is the laser power. The ENBW is the ENBW which should be 1,250 Hz, when the time constant and filter slope of the LIA are set to 100 μs and 12 dB, respectively. Hence, the NNEA = $2.45 \times 10^{-9}$ cm$^{-1}$ W Hz$^{-1/2}$ can be calculated if $\alpha_{\min} = 9.3 \times 10^{-6}$ cm$^{-1}$ and $P = 13.04$ mW.

**Table 1 | Detection limits of BF-QEPA-based sensor with three different kinds of semiconductor lasers.**

| Laser type | Frequency (cm$^{-1}$) | Gas type | Technique | Pressure (Torr) | Integration time (ms) | $f_t/f_0$ (Hz) | $Q_t/Q_0$ | NEC (p.p.m.) | NNEA (cm$^{-1}$ W Hz$^{-1/2}$) |
|---|---|---|---|---|---|---|---|---|---|
| DFB-DL | 7,306.75 | $H_2O$ | QEPAS | 760 | 300 | – | – | 0.41 | $1.5 \times 10^{-8}$ |
| | | | BF-QEPAS | | 0.1 | 32,760 / 32,760 | 1,848/1,846 | 0.059 | $2.45 \times 10^{-9}$ |
| DFB-QCL | 2,190.02 | CO | QEPAS | 700 | 300 | – | – | 0.011 | $2.4 \times 10^{-8}$ |
| | | | BF-QEPAS | | 3 | 32,755 / 32,755 | 1,851/1,846 | 0.01 | $2.3 \times 10^{-8}$ |
| DFB-ICL | 2,778.64 | $CH_4$ | QEPAS | 700 | 300 | – | – | 55.57 | $1.81 \times 10^{-8}$ |
| | | | BF-QEPAS | | 3 | 32,757 / 32,757 | 1,817/1,823 | 40.75 | $1.3 \times 10^{-8}$ |

BF, beat frequency; DFB-DL, distributed feedback diode laser; DFB-ICL, DFB interband cascade laser; DFB-QCL, DFB quantum cascade laser; NEC, noise-equivalent concentration; NNEA, normalized noise equivalent absorption coefficient; QEPAS, quartz-enhanced photoacoustic spectroscopy.
NEC for available laser power and a 1 s integration time.
NNEA data were calculated from NEC data and HITRAN database (Methods section).

**QTF response profile as a function of frequency.** The apparatus mentioned in the experimental apparatus section was used to measure the QTF response profile as a function of frequency by measuring the peak value of the conventional 2$f$ QEPAS signal at room temperature and atmosphere pressure with 2.5% water vapour. The time constant and filter slope of the LIA were set to 300 ms and 12 dB. The wavelength scanning rate was $0.18 \, \mathrm{cm}^{-1} \mathrm{s}^{-1}$.

**Data availability.** The authors declare that all data supporting the findings of this study can be found within the paper and its Supplementary Information Files. Additional data supporting the findings of this study are available from the corresponding author (L.D.) upon reasonable request.

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

## Acknowledgements

This material is based upon work supported by National Natural Science Foundation of China (Grants #61622503, 61575113, 61275213). Frank K. Tittel acknowledges Grant C-0586 by the Welch Foundation.

## Author contributions

H.W. and L.D. designed the experiments. F.K.T. supervised the project. H.W., Y.Y., W.M., L.Z., H.Z. performed the reported experiments. L.D., W.Y., L.X. and S.J. conducted the experimental analysis. H.W., L.D. and F.K.T. prepared the manuscript.

## Additional information

**Competing interests:** The authors declare no competing financial interests.

