## [Peer Review File · Nature Communications]

Reviewers' comments:

Reviewer #1 (Remarks to the Author):

In my opinion manuscript adds some value to the subject of laser based trace gas detection by presenting a stimulating new QEPAS method. QEPAS has been around for a long time but due to difficulties in rugged calibration and matrix interference it has found not many practical applications yet.

This paper provides a new, calibration free technique with direct access to the actual Q-factor of the tuning fork employed. This is a very important new development and might turn out truly relevant for practical implementation of this technology for trace gas sensing in ultra-low gas volumes.

The paper is extremely well written and provides a solid theoretical as well as convincing experimental results. Therefore the manuscript deserves to be published after clarifying some major points:

Conventional QEPAS employs 2f wavelength modulation among others because it provides high selectivity together with background free signals. These features are very important for a wide range real-world applications. What about selectivity and background signals within the presented technique? It seems that a similar degree in selectivity can not be achieved using the proposed pulsed modulation scheme. Please discuss this important issue and highlight differences to conventional QEPAS in the Discussion section in detail.

Page 4, lines 67-68: High selectivity is also an important requirement for on-line monitoring. Please add this requirement and explain how it can be reached by the new method (In conventional QEPAS this is achieved by the 2f-wavelength modulation technique).

Line 83: how long does such a calibration process take (ms to sec)??

Line 88: Rapid QEPAS measurements with a time constant of 15 ms have been shown, inserted citation:

A. A. Kosterev, P. R. Buerki, L. Dong, M. Reed, T. Day, and F. K. Tittel, "QEPAS detector for rapid spectral measurements", *Applied Physics B: Lasers and Optics*, vol. 100, no. 1, 2010.

Line 127: what is rapid? Give a number. What is the repetition rate, pulse duration for a single QTF excitation?

Fig. 4: why are slopes of 1st and 3rd harmonic of the same sign? They should be opposite; please comment on this.

Page 18: It is strongly suggested to provide a NNEA value for an integration time of 1 second, so that the measurement is directly comparable to conventional QEPAS.

Discussion:

Table 1: Laser types: DFB is no laser type: should be diode laser; Most likely also the employed QCL and ICL were DFB-QCL and DFB-ICL, correct? Please provide more information about these lasers sources, type, power, ... ; Please discuss why the integration times for the BF-QEPAS measurements are different (0.1 and 3 ms). Why is the Q-factor that low, in the intro it was mentioned that it is up to 15000.

Line 337: Please discuss in detail why the NEC value is higher for the BF-QEPAS method than for the conventional method, even though the 1st harmonic was used. What about signal generation process differences between pulsed and CW modulation?

Line 334: the opposite is shown in the paper: the new technique does not show improved sensitivity; it shows fast data acquisition but worse sensitivity compared to conventional QEPAS systems. Please clarify this important issue.

Further minor points to be clarified:

Page 2, line 27: QTF: introduce abbreviation, line 28: only QTF

Line 69: the commercial QTFs have a resonance frequency of 33 kHz (when omitting digits).

Line 70: in case of no varying gas matrix and temperature the system has only to be calibrated once.

Line 74: change "The modulation frequency of the laser beam" to "the modulation of the laser frequency"

Fig 1: CW acoustic wave: this description is redundant. It is suggested to use "acoustic wave"; and to use "acoustic pulse" instead of "pulsed acoustic wave"

Line 125: why calculated?

Line 126: what is a pulsed acoustic wave? Either it is a wave or a pulse

Fig. 2: In the legend it is stated that three different semiconductor lasers were employed; However, the schematic only shows only one laser source, also within the section experimental apparatus only one laser source is described; please clarify! Please also explain the abbreviations ICL and QCL. These lasers are not described in this section!

Line 174: This sentence is confusing. Why provides a long interaction time a slowly varying acoustic wave??

Line 176: and throughout the manuscript. The wording "CW acoustic wave" is not clear at all. Why the description "continuous WAVE (CW) acoustic WAVE"??

Line 221: why/how is the absorption line locked?

Reviewer #2 (Remarks to the Author):

The manuscript „ Calibration-free and fast quartz-enhanced photoacoustic spectroscopy based on a beat frequency effect for continuous trace gas monitoring" by Wu et al. describes a new method for photoacoustic measurement where the transient acoustic wave is used for excitation rather than an equilibrium state. The article is clearly written and the statements are supported by sound measurements and clear figures. I recommend the manuscript for publication; however below are some remarks and suggestions to further improve it prior to final submission:

The demands for on-line monitoring (lines 67/68) should not only include compact size but also a small gas volume.

Line 74: modulation frequency of the laser radiation, not laser beam

Line 176/177: you calculate the detection bandwidth with $1/(4T)$; however the manual of the SRS830 gives this formula only for 6dB/octave. For 12dB/octave (what is what you used later and presumably also here), you should use $1/(8T)$. This remark holds throughout the manuscript and you should check the whole text and correct the bandwidths and the NNEA accordingly.

Line 187: How can you reduce the background noise by reduction of the averaging time? Do you mean maybe a trade-off between a sufficient bandwidth and still not too high background noise level?

Figure 3: why does the oscillation in figure 3c does start instantly? The beginning at the time when the absorption line is reached by tuning the ramp as in the experimental data in figure 3f is much more plausible.

When you discuss the graph in figure 3b, you describe the exponentially decaying tail. What happens afterwards? What is the rising part afterwards?

Line 201-203: of course the interaction time t_a in the simulation is connected with the scan-rate in the experiment. However, for me it is not clear how you adjusted the time/rate to each other? Please explain! It would also be interesting, how the relaxation time is accounted for in the simulation when a not so rapid relaxer as water vapor is simulated.

You emphasize that you can determine the resonance frequency with the method and thus avoid frequent recalibration. Please discuss which deviation is acceptable – you start in the simulation with a value of 32755Hz (line 174) and retrieve out of the simulated curve a value of 32762Hz, which is a deviation of 7 Hz, which is, given the sharp peak of the resonance curve of the actual TF a lot. Please discuss.

Line 221: should it be "located" instead of "locked"?

Figure 5: did you also simulate these dependencies?

Line 374: mW

Supplementary Figure 3: 10ms, 30ms and 100ms cannot be distinguished. I would find it more instructive, to add more graphs for values between 100 μ s and 10 μ s as there should be the trade-off between signal strength and noise. Also it would maybe be instructive, if you would plot Signal/noise vs. time constant to determine the best value.

Supplementary Figure 4: please discuss the different shapes. The figure captions in the main text are much better than in the supplement.

Supplement line 71: delete one "the"

Supplement figure 7: figure and figure caption are contradictive: in the figure, a scan rate of 50 cm-1/s is given, while the caption gives 55cm-1/s. Furthermore, line 84 states ICL while line 92 states QCL?

Line 103: what does "for this purpose" relate to?

Beat frequency signal analysis: could you also use the different slopes for a more precise evaluation of the concentration?

Line 140: add a space before CO.

Line 147/148: sentence is hard to read. Use one sentence for CO and one for CH4.

Line 171-174: this is a crude method to approximate the derivative – please discuss why it is sufficient/appropriate.

Since you are using the transient response, there is no standing acoustic wave that could build up within the acoustic resonator. I wonder why you still use it? Please comment on this.

References: there are quite a few references cited that hardly touch what is discussed within this paper (other QEPAS-related techniques). Check if you need all of them.

Response to reviewers' comments

Please Note: referees' comments are in black; our comments are in *red italic*; the original paper text in red and revised text in blue.

Reviewer #1 (Remarks to the Author):

In my opinion manuscript adds some value to the subject of laser based trace gas detection by presenting a stimulating new QEPAS method. QEPAS has been around for a long time but due to difficulties in rugged calibration and matrix interference it has found not many practical applications yet.

This paper provides a new, calibration free technique with direct access to the actual Q-factor of the tuning fork employed. This is a very important new development and might turn out truly relevant for practical implementation of this technology for trace gas sensing in ultra-low gas volumes.

The paper is extremely well written and provides a solid theory as well as convincing experimental results. Therefore the manuscript deserves to be published after clarifying some major points:

Conventional QEPAS employs $2f$ wavelength modulation among others because it provides high selectivity together with background free signals. These features are very important for a wide range real-world application. What about selectivity and background signals within the presented technique? It seems that a similar degree in selectivity can not be achieved using the proposed pulsed modulation scheme. Please discuss this important issue and highlight differences to conventional QEPAS in the Discussion section in detail.

With a conventional QEPAS based sensor, the high selectivity results from the selection of the interference-free absorption line of a target gas as well as the background suppression of the $2f$ wavelength modulation technique. The conventional QEPAS technique modulates and demodulates the QEPAS signal at every wavelength point within the wavelength scanning range. Therefore the x-axis of the conventional $2f$ QEPAS signal corresponds to 'wavelength'. 'Wavelength' is sometimes replaced with 'time' since the laser wavelength changes linearly over time.

In the new BF-QEPAS technique, the condition is completely different. The signal detection is divided into two stages. In the first stage, a rapid wavelength scanning crossing the target absorption line was performed, which results in an acoustic pulse. The acoustic pulse pushes the prongs of the QTF to vibrate in a short period of time. In the second stage, the laser wavelength scanning is terminated. The QTF prong changes to a free vibration mode and meantime the harmonic signals are demodulated. We described the details in the section of Theory of BF-QEPAS (page 7 and Fig. 2 (b)). Therefore the x-axis of the BF-QEPAS signal just corresponds to 'time', not 'wavelength'. It is not proper to assess the selectivity by directly

comparing the conventional QEPAS signal with the BF-QEPAS signal. In fact, after the interference-free absorption line is selected, the selectivity of the BF-QEPAS technique is mainly determined in the first stage when a rapid wavelength scanning was carried out crossing the target absorption line. The target absorption line produces an only acoustic pulse, which ensures the high selectivity.

Furthermore, in the second stage, the QTF vibrating freely at the resonant frequency possesses a high Q-factor, which effectively suppresses the background noise from the acoustic pulse with a non-resonant frequency in the first stage. The QTF just remains the intensity information of the acoustic pulse. As a result, BF-QEPAS technique is dominated by the QTF thermal noise whether 1f or 2f detection is used.

We also added the corresponding sentence in the Discussion section on page 20:

“The BF-QEPAS technique similar to the conventional 2f wavelength modulation based QEPAS technique provides high selectivity together with background-free signals. Its selectivity is mainly determined by an initial rapid wavelength scanning crossing the target absorption line. The target absorption line produces an only acoustic pulse to push the QTF prongs to vibrate, ensuring high selectivity. Subsequently the high Q-factor of the QTF vibrating freely at its resonant frequency effectively suppresses the background noise from the non-resonant acoustic pulse. Only the intensity information of the acoustic pulse remains in the BF-QEPAS signal. As a result, the BF-QEPAS technique is dominated by the QTF thermal noise using either 1f or 2f detection.”

Page 4, lines 67-68: High selectivity is also an important requirement for on-line monitoring. Please add this requirement and explain how it can be reached by the new method (In conventional QEPAS this is achieved by the 2f-wavelength modulation technique).

The question is the same as the first one. We made a detailed answer above.

We also added the phrase in the corresponding sentence on page 4, paragraph 1:

“Requirements for on-line monitoring of trace gases are uninterrupted operation, fast response, high selectivity, high detection sensitivity and compact size with a small gas cell.”

Line 83: how long does such a calibration process take (ms to sec)??

*As described in Ref. 8 and the papers which were published by Prof. Dong (Dong. L. et al. Compact QEPAS sensor for trace methane and ammonia detection in impure hydrogen. Appl. Phys. B **107**, 459 (2012)) and Prof. Kosterev (Kosterev. A. A. et al. Applications of quartz tuning forks in spectroscopic gas sensing. Rev. Sci. Instrum. **76**, 043105 (2005)), the electric excitation method is completed by applying an AC voltage to the QTF and scanning frequency of the applied voltage. The response frequency and the Q-factor of the QTF can be calculated based on the complete QTF frequency response profile. The time required to complete such a calibration process depends on the scan range and scan step. With a bare QTF, the narrow frequency response profile results in an ~30 s calibration time, and the calibration time can*

extend to ~90 s for a QTF coupled with an acoustic micro-resonator (see Fig. 3 in the Ref. 35) as its low Q-factor results in a wide frequency response profile.

To clarify this, we replaced the sentence on page 4, paragraph 1:

“Such a calibration process interrupts a continuous QEPAS measurement...”

with the following one:

“Such a calibration process, which usually requires ~ 90 s to complete, interrupts a continuous QEPAS measurement.”

Line 88: Rapid QEPAS measurements with a time constant of 15 ms have been shown, inserted citation: A. A. Kosterev, P. R. Buerki, L. Dong, M. Reed, T. Day, and F. K. Tittel, "QEPAS detector for rapid spectral measurements", Applied Physics B: Lasers and Optics, vol. 100, no. 1, 2010.

The time constant of the rapid QEPAS measurement, 15 ms, mentioned in Ref. 26 was obtained using the equation: $\tau=Q/\pi \cdot f_0$. This is a theoretically estimated value and there was no corresponding experimental result reported in this article. In fact, the experimentally obtained time constant of the rapid spectral measurements studied in Ref. 26 was 39 ms. Furthermore, as mentioned by the authors of that paper, the per-point measurement time is ~3 times longer than the QTF response time. Therefore the measurement cycle of the rapid spectral measurements should be ~117 ms which is longer than that the BF-QEPAS required.

Line 127: what is rapid? Give a number. What is the repetition rate, pulse duration for a single QTF excitation?

As mentioned in the captions of Fig. 4 and Fig. 7, we use wavelength-scanning rate to reflect the rapid spectral measurements processes.

To clarify this, we replaced the sentence on page 7, paragraph 2:

“In BF-QEPAS a pulsed acoustic wave induced by the target gas absorption is generated as a result of rapid current scanning, which causes...”

with the following one:

“In BF-QEPAS an acoustic pulse induced by the target gas absorption is generated as a result of rapid wavelength scanning ($>30 \text{ cm}^{-1} \cdot \text{s}^{-1}$), which causes...”

To clarify the repetition rate of the scanning signal, we added the following sentence on Page 8:

“and then remained at a constant value to complete the induced BF-QEPAS signal detection. The scanning cycle is the sum of the scanning time and the waiting time. The repetition rate of the ramp signal is 12 Hz. The waiting time of a single scanning cycle is 0.05 s”

Fig. 4: why are slopes of 1st and 3rd harmonic of the same sign? They should be opposite; please comment on this.

The signal demodulated by a lock-in amplifier (LIA) has two detection channels corresponding to the in-phase and quadrature components. To obtain the maximize signal amplitude, we usually adjust the LIA phase to make the signal amplitude from one of the two channels close to zero and detect the signal from the other channel. However in the process of a phase adjustment, we focused on the amplitude and neglected the phase relationship between 1st and 3rd harmonics. Many thanks for the reviewer's question. It is true that the slopes of 1st and 3rd harmonic should be opposite. We replotted Fig. 4 to avoid confusion.

Page 18: It is strongly suggested to provide a NNEA value for an integration time of 1 second, so that the measurement is directly comparable to conventional QEPAS.

The normalized noise equivalent absorption (NNEA) coefficient is defined as the minimum optical absorption coefficient ($S=N$) multiplied by the optical excitation power and divided by the detector bandwidth. Therefore, the NNEA is independent of the laser power, absorption line of the target gas as well as the integration time. This is the reason why the NNEA value is often used to characterize the trace gas detection performance of a photoacoustic absorbance detection module.

Instead, we normalized the noise-equivalent concentration (NEC) to an 1-s integration time, as shown in Table. 1, in order for the readers to directly compare the BF-QEPAS based sensors with QEPAS based sensors.

We replaced the sentence on page 18, paragraph 2:

“The noise-equivalent concentration (NEC) of the BF-QEPAS-based sensor was estimated...”

With the following one

“The noise-equivalent concentration (NEC) of the BF-QEPAS-based sensor for 0.1-ms integration time was estimated ...”

We added the relevant sentences on page 19, paragraph 3:

“The NEC was normalized to a 1-s integration time to facilitate inter-comparison.”

The relevant sentences were also added in Table. 1

“NEC for available laser power and an 1-s integration time”

In addition, we replaced the sentence on page 19, paragraph 3:

“The BF-QEPAS technique has a higher NEC value than the conventional QEPAS technique for all three laser sources and the BF-QEPAS has an improved NNEA value than the conventional QEPAS technique.”

with the following one:

“The BF-QEPAS technique has an obviously lower NEC and NNEA value than the conventional QEPAS technique for the fast relaxing molecule (H₂O). However, for the slow relaxing molecules (CH₄ and CO), the NEC and NNEA for BF-QEPAS technique are somewhat better than the conventional QEPAS technique.”

Discussion:

Table 1: Laser types: DFB is no laser type: should be diode laser; Most likely also the employed QCL and ICL were DFB-QCL and DFB-ICL, correct?

We replaced the DFB, QCL, ICL in the manuscript and supplementary with the DFB-DL, DFB-QCL, DFB-ICL.

Please provide more information about these lasers sources, type, power, ...

In fact, all information regarding the type, power, etc. of these laser sources can be found in manuscript text. The information about the DFB-DL laser was described on page 7, last paragraph:

“A 1368.7 nm distributed feedback diode laser (DFB-DL) (NTT Electronics, Inc. Model NLK1E5E1AA) was employed as the excitation source.”

and on page 22, paragraph 1:

“...and P = 13.04 mw.”

The information about the DFB-QCL laser was described in the caption of the Supplementary Figure 6:

“The DFB-QCL (AdTech Optics, Inc. Model HHL-14-32) temperature and current were

selected to target a CO absorption line located at $2,190.02 \text{ cm}^{-1}$ with a 26.5 mW output power and line-strength of $2.915 \times 10^{-19} \text{ cm} \cdot \text{mol}^{-1}$.”

The information about the DFB-ICL laser was described in the caption of the Supplementary Figure 7:

“The DFB-ICL (Nanoplus Nanosystems and Technologies GmbH, S/N: 1485/25-19) temperature and current was selected to target a CH₄ absorption line located at $2,778.64 \text{ cm}^{-1}$ with a 2.3 mW output power and line-strength of $5.241 \times 10^{-22} \text{ cm} \cdot \text{mol}^{-1}$.”

Please discuss why the integration times for the BF-QEPAS measurements are different (0.1 and 3 ms).

The Photoacoustic spectroscopy (PAS) signal depends on the vibration-translation (V-T) relaxation rates of target gases. It is well known that the CO and CH₄ have slower V-T relaxation rates than H₂O, which implies that the PAS signals of CO and CH₄ are weaker. Therefore, a narrower detection bandwidth was required to improve the signal-to-noise ratio (SNR) when the BF-QEPAS signals of the CO and CH₄ were detected. The 0.1 ms for H₂O detection and 3 ms for CO and CH₄ detection were the experimental optimized values, which verified that the line shape of the BF-QEPAS signal was not distorted and maintained the narrow bandwidth required for noise reduction.

To clarify this fact, we added the following sentence in the supplement on page 11:

“A 3-ms integration time was used to improve the SNR when the BF-QEPAS signals of the CO and CH₄ were detected.”

We also added the replaced the sentence in the manuscript on page 20, last paragraph:

“The results show that this new technique is capable of providing improved detection sensitivity and reduced data acquisition times than those reported for conventional QEPAS.”

with the following sentence:

“The results show that this new technique is capable of providing reduced data acquisition times and improved detection sensitivity than those reported for conventional QEPAS, especially for fast relaxing molecules.”

Why is the Q-factor that low, in the intro it was mentioned that it is up to 15000.

The Q-factor of a bare QTF in atmospheric pressure is usually more than ten thousand, and it decreases to a few thousand when the QTF is well coupled with the acoustic micro-resonator (AmR). Furthermore, as mentioned on page 4, the Q-factor is subject to the fabrication process and also depends on the operating environment. For the bare QTF used in our experiment, its Q-factor was 13,476 in atmospheric pressure (760 Torr). When the QTF was coupled with the optimized AmR, the value of the Q-factor decreased to 2,203. The Q-factor dropped to 1,850 after the QTF was immersed in the target gas with high humidity for more

than one hour. The variation of the Q-factor shows the necessity and importance of the real-time measurement of the Q-factor.

In order to clarify it, we added a sentence on page 21, paragraph 1:

“In this case, the Q-factor of the QTF decreased to 1,850 as a result of coupling between the QTF and the AmRs^{21,35}.”

Line 337: Please discuss in detail why the NEC value is higher for the BF-QEPAS method than for the conventional method, even though the 1st harmonic was used.

The BF-QEPAS method has higher NEC values because it has a 2-3 orders of magnitude wider detection bandwidth. According to the reviewer’s suggestion, we normalized the NEC values to an 1-s integration time to facilitate a comparison. Thus the BF-QEPAS method shows better NEC values.

What about signal generation process differences between pulsed and CW modulation?

The BF-QEPAS technique also uses CW modulation. There is not difference in the wavelength modulation method.

To avoid confusion, we replaced the sentence on page 8:

“The sinusoidal AC component, generated from ...”

With the following one:

“The continuous sinusoidal AC component, generated from ...”

The difference between the two techniques is the signal generation process, as described in the caption of Fig. 1:

“Unlike conventional QEPAS, the modulation frequency f of the laser in the BF-QEPAS technique is shifted from the QTF resonance frequency. The laser wavelength is rapidly scanned with respect to the QTF response time.”

and on page 7, paragraph 2:

“In conventional QEPAS, a slowly varying continuous acoustic wave causes forced vibrations of the QTF. The transient response of the QTF is neglected and only the steady state behavior is taken into account due to the long averaging time (>300 ms) used during which the transient response is averaged to be zero. In BF-QEPAS an acoustic pulse induced by the target gas absorption is generated as a result of rapid wavelength scanning (>30 $\text{cm}^{-1} \cdot \text{s}^{-1}$), which causes the prongs of the QTF to vibrate in a short period of time. Subsequently the QTF prong changes to a free vibration mode after the acoustic pulse wave terminates rather than the continuous forced vibrations caused by a continuous acoustic wave as in conventional QEPAS. The vibration energy will be dissipated via extrinsic and intrinsic QTF loss mechanisms³⁴. At this point, the QTF is vibrating at its resonance frequency and not at the laser modulation frequency. The QTF signal is

demodulated at the laser modulation frequency f . When the averaging time is short enough (<100 ms) to provide a sufficient system detection bandwidth, a beat frequency signal with an exponential decay envelope is generated from the QTF transient response.”

In addition, the details regarding the two signal generation processes was also described in the answer of the first issue.

Line 334: the opposite is shown in the paper: the new technique does not show improved sensitivity; it shows fast data acquisition but worse sensitivity compared to conventional QEPAS systems. Please clarify this important issue.

The reason why the new technique shows higher NECs is because we did not normalize the NECs to an 1-s integration time. It is not fair to compare the NECs with different integration times. After NEC normalization, the new technique shows improved sensitivity, especially for fast relaxing molecules, as shown in Table 1.

Further minor points to be clarified:

Page 2, line 27: QTF: introduce abbreviation, line 28: only QTF

Done

Line 69: the commercial QTFs have a resonance frequency of 33 kHz (when omitting digits).

Done

Line 70: in case of no varying gas matrix and temperature the system has only to be calibrated once.

It is theoretically correct except for the obsolescence problem of the QTF. However, it is not known whether the gas matrix has changed or not. This is what the gas sensor will tell us.

Line 74: change “The modulation frequency of the laser beam” to “the modulation of the laser frequency”

We replaced the sentence on page 4, paragraph 1:

“The modulation frequency of the laser beam (f) must accurately match with the QTF resonance frequency to obtain the highest signal amplitude.”

with the following one:

“The modulation of the laser radiation frequency (f) must accurately match with the QTF resonance frequency to obtain the highest signal amplitude.”

Fig 1: CW acoustic wave: this description is redundant. It is suggested to use “acoustic wave” ; and to use “acoustic pulse” instead of “pulsed acoustic wave”

Done

Line 125: why calculated?

We wanted to convey the fact that only the steady state behavior should be taken into account in conventional QEPAS. To clarify this, we replaced the word on page 7, paragraph 2:

“calculated”

With the phrase:

“taken into account”

Line 126: what is a pulsed acoustic wave? Either it is a wave or a pulse

We replaced the “pulsed acoustic wave” in the manuscript with “acoustic pulse”. And the “quasi-pulsed acoustic wave” was replaced with “quasi-pulsed acoustic signal”.

Fig. 2: In the legend it is stated that three different semiconductor lasers were employed; However, the schematic only shows only one laser source, also within the section experimental apparatus only one laser source is described; please clarify!

As the three different semiconductor lasers were employed in this system sequentially and they were driven via the same equipment, we use the phrase “laser excitation source” to replace the three laser source for brevity. To clarify it, we replaced the sentence on the caption of Fig. 2:

“Three different semiconductor lasers were employed in this system as the excitation sources sequentially. Optical lenses were used to collimate the ICL and QCL laser beams. A fiber-coupled collimator ensures that the collimated DFB diode laser beam passes through the ADM without touching the QTF prongs.”

With the phrase:

“Three different semiconductor lasers, DFB-DL, DFB quantum cascade laser (DFB-QCL) and DFB interband cascade laser (DFB-ICL), were employed in this system as the excitation sources sequentially. A fiber-coupled collimator ensures that the collimated DFB-DL beam passes through the ADM without touching the QTF prongs. Optical lenses were used to collimate the DFB-ICL and DFB-QCL laser beams. The details about the experiments, in which the DFB-QCL and DFB-ICL were equipped as the excitation source, were described in the Supplementary section.”

Please also explain the abbreviations ICL and QCL. These lasers are not described in this section!

Done

Line 174: This sentence is confusing. Why provides a long interaction time a slowly varying acoustic wave??

The frequency of the acoustic wave is 32,760 Hz which is a constant. "A slowly varying CW acoustic wave" means "a continuous acoustic wave with a slow amplitude variation"

To clarify it, we replaced the sentence on page 10:

"...the interaction time t_a between the acoustic wave and the QTF was long enough ($t_a = 10$ s) to provide a slowly varying CW acoustic wave."

with the following one:

"...the interaction time t_a between the acoustic wave and the QTF was long enough ($t_a = 10$ s) to provide a continuous acoustic wave with a slow amplitude variation."

Line 176: and throughout the manuscript. The wording "CW acoustic wave" is not clear at all. Why the description "continuous WAVE (CW) acoustic WAVE" ??

We replaced the "CW acoustic wave" with the "continuous acoustic wave" throughout our manuscript.

Line 221: why/how is the absorption line locked?

We have replaced the word "locked" on page 12, last paragraph with "located".

Reviewer #2 (Remarks to the Author):

The manuscript "Calibration-free and fast quartz-enhanced photoacoustic spectroscopy based on a beat frequency effect for continuous trace gas monitoring" by Wu et al. describes a new method for photoacoustic measurement where the transient acoustic wave is used for excitation rather than an equilibrium state. The article is clearly written and the statements are supported by sound measurements and clear figures. I recommend the manuscript for publication; however below are some remarks and suggestions to further improve it prior to final submission:

The demands for on-line monitoring (lines 67/68) should not only include compact size but also a small gas volume.

We replaced the sentence on page 4, paragraph 1:

“Requirements for on-line monitoring of trace gases are uninterrupted operation, fast response, compact size and high sensitivity.”

with the following one:

“Requirements for on-line monitoring of trace gases are uninterrupted operation, fast response, high selectivity, high sensitivity and compact size with a small gas cell.”

Line 74: modulation frequency of the laser radiation, not laser beam

We replaced the sentence on page 4, paragraph 1:

“The modulation frequency of the laser beam (f) must accurately match with the QTF resonance frequency to obtain the highest signal amplitude.”

with the following one:

“The modulation of the laser radiation frequency (f) must accurately match with the QTF resonance frequency to obtain the highest signal amplitude.”

Line 176/177: you calculate the detection bandwidth with $1/(4T)$; however the manual of the SRS830 gives this formula only for 6dB/octave. For 12dB/octave (what is what you used later and presumably also here), you should use $1/(8T)$. This remark holds throughout the manuscript and you should check the whole text and correct the bandwidths and the NNEA accordingly.

This is true that, for the SR830 Lock-in amplifier (LIA), the detection bandwidth should be $1/8T$ when the 12dB/octave filter slope was selected, according to the LIA manual. But it should be noted that there are two detection channels in the LIA and the detection bandwidth in the manual is just for one detection channel.

As mentioned on page 16, the total fundamental noise includes the QTF thermal noise $\sqrt{\langle V_{N-R}^2 \rangle}$ and the feedback resistor noise $\sqrt{\langle V_{N-R_g}^2 \rangle}$. The noise power density of $\sqrt{\langle V_{N-R}^2 \rangle}$ should be integrated over the QTF resonant curve, which is narrower than the LIA detection bandwidth. So the $\sqrt{\langle V_{N-R}^2 \rangle}$ is independent from the LIA detection bandwidth and is the noise power density before assigned to each detection channel. The $\sqrt{\langle V_{N-R_g}^2 \rangle}$ is a function of the LIA detection bandwidth Δf_{dec} . In order to obtain the noise sum of the thermal noise and the feedback resistor noise, the Δf_{dec} should use the total detection bandwidth, rather than

the detection bandwidth of a single detection channel. This is why the 1/4T was used instead of 1/8T. After that, the total noise can be assigned to each channel using a 1/√2 coefficient.

The total noise should be expressed in equation 8 as:

$$\sqrt{\langle V_N^2 \rangle} = \frac{1}{\sqrt{2}} \cdot \left(\sqrt{\langle V_{N-R}^2 \rangle} + \sqrt{\langle V_{N-R_g}^2 \rangle} \right) = \frac{1}{\sqrt{2}} \cdot \left(R_g \cdot \sqrt{\frac{2\pi \cdot k_B \cdot T \cdot f_0}{R \cdot Q}} + \sqrt{4k_B \cdot T \cdot R_g \cdot \Delta f_{\text{det}}} \right)$$

We replaced the sentence on page 16, paragraph 2:

“...and Δf_{det} is the detection bandwidth of the system.”

With the following one:

“...and Δf_{det} is the total detection bandwidth of the system for the two LIA detection channels.”

We also added the corresponding sentence on page 17, paragraph 1:

“The $1/\sqrt{2}$ coefficient reflects the fact that the noise is calculated only for one detection channel.”

To avoid confusion, we used the 1/8T when describing the detection bandwidth and revised them throughout the manuscript, accordingly. The “2,500 Hz” and “0.833 Hz” were changed to “1,250 Hz” and “0.417 Hz”.

The NNEA in Table 1 was also corrected.

Line 187: How can you reduce the background noise by reduction of the averaging time? Do you mean maybe a trade-off between a sufficient bandwidth and still not too high background noise level?

We mean a trade-off between a sufficient bandwidth and still not too high background noise level.

To clarify this, we replaced the sentence on page 10:

“...and to reduce the background noise.”

with

“...and to maintain efficient background noise suppression.”

Figure 3: why does the oscillation in figure 3c does start instantly? The beginning at the time when the absorption line is reached by tuning the ramp as in the experimental data in figure 3f is much more plausible.

In order to obtain the useful simulation signal, the program for simulating the QTF output

signal for Fig. 3c employed an absorption line without a large flat wings on both sides as the driving force $U(t)$ in equation (2). This is why the oscillation in Fig. 3c starts instantly. However the absence of the large flat wings does not distort the BF-QEPAS signal.

When you discuss the graph in figure 3b, you describe the exponentially decaying tail. What happens afterwards? What is the rising part afterwards?

With direct absorption spectroscopy, an exponentially decaying tail can be observed as shown in Fig. 4c of Ref. 26 as all values are greater than zero. But the 1st harmonic signal has a positive peak and a negative valley. The rising part that the reviewer mentioned is the exponentially decaying tail of the negative valley.

To clarify this, we replaced the sentence on page 10:

“An exponentially decaying tail was observed when the LIA detection bandwidth ...”

with the following one:

“Two exponentially decaying tails, related to the peak and the valley of the 1st harmonic signal were observed when the LIA detection bandwidth...”

Line 201-203: of course the interaction time t_a in the simulation is connected with the scan-rate in the experiment. However, for me it is not clear how you adjusted the time/rate to each other? Please explain!

We estimated the interaction time t_a using the ratio of the absorption line width to the wavelength scanning rate. The absorption line width can be precisely calculated based on HITRAN line parameters and environment parameters (pressure, temperature and modulation depth). Therefore, the t_a in the simulation program can be obtained according to the wavelength scanning rate used in our experiments.

To clarify this point, we added the corresponding sentence in the caption of Fig. 3:

“The value of t_a was estimated by using the ratio of the absorption line width to the wavelength scanning rate.”

It would also be interesting, how the relaxation time is accounted for in the simulation when a not so rapid relaxer as water vapor is simulated.

The current theoretical model of the BF-QEPAS has not considered the relaxation time of target gases. This will be our future work.

You emphasize that you can determine the resonance frequency with the method and thus avoid frequent recalibration. Please discuss which deviation is acceptable – you start in the simulation with a value of 32755Hz (line 174) and retrieve out of the simulated curve a value

of 32762Hz, which is a deviation of 7 Hz, which is, given the sharp peak of the resonance curve of the actual TF a lot. Please discuss.

Many thanks for this reviewer's question. We accidentally mixed up the QTF resonance frequencies of two experiments when we prepared the manuscript. The resonant frequency of 32,755Hz is from the DFB-ICL experiment, as shown in Table 1. For the current DFB-DL experiment, the resonant frequency is 32,760 Hz.

To correct this mistake, we replaced the sentence on page 10:

“In Fig. 3a, a QTF resonance frequency, $f_0 = 32,755$ Hz, was employed as the laser modulation frequency and ...”

with the following one:

“In Fig. 3a, a QTF resonance frequency, $f_0 = 32,760$ Hz, was employed as the laser modulation frequency and ...”

And the correspond sentence on the caption of Fig. 4:

“The laser modulation frequencies for standard 1st, 2nd and 3rd harmonic were 32,755 Hz, 16,377.5 Hz and 10,918.3 Hz, while they...”

was replaced with the following one:

“The laser modulation frequencies for standard 1st, 2nd and 3rd harmonic were 32,760 Hz, 16,380 Hz and 10,920 Hz, while they...”

The acceptable deviation of the resonant frequency is ± 10 Hz because with such a deviation, the fluctuation of the signal amplitude can be controlled within $\pm 0.5\%$, according to Fig. 5.

Line 221: should it be “located” instead of “locked” ?

Done

Figure 5: did you also simulate these dependencies?

We did not simulate these dependencies in Fig. 5. These data were obtained from the experiments.

Line 374: mW

Done

Supplementary Figure 3: 10ms, 30ms and 100ms cannot be distinguished. I would find it more instructive, to add more graphs for values between $100\mu\text{s}$ and $10\mu\text{s}$ as there should be the trade-off between signal strength and noise.

The BF-QEPAS signals for 10 ms, 30 ms and 100 ms were plotted in the Response Fig. 1a. The results show that the LIA detection bandwidth was too narrow to detect the perfect BF-QEPAS signal. We did not add this figure and this comment in the revised manuscript, since the BF-QEPAS based sensor should not be operated in such cases.

Response Fig. 1 BF-QEPAS signal for different LIA time constant.

For the LIA (SR830), there is a setting of 30 μs between 10 μs and 100 μs. We added the BF-QEPAS signal for 30 μs in Supplementary Figure 3, according to the reviewer's suggestion. The BF-QEPAS signals for 30 μs and 100 μs are also shown in the Response Fig. 1b. The BF-QEPAS signal for 30 μs has a similar signal amplitude but with a higher noise level compared with the BF-QEPAS signal for 100 μs.

Also it would maybe be instructive, if you would plot Signal/noise vs. time constant to determine the best value.

This is an excellent suggestion. Normalized SNR value as a function of the LIA integration time was added to Supplementary Figure 3.

The relevant sentences were also added into the Supplementary Figure 3 caption:

Supplementary Figure 3: (a)-(c) BF-QEPAS signal for different LIA time constants. (d) Normalized SNR value for different LIA time constants. A DFB diode laser emitting at 1.368 μm was used as the excitation source. The modulation frequency and depth of the wavelength was 32960 Hz and 1 cm^{-1} , respectively. The filter slope of the LIA was set at 12 dB. The wavelength was scanned at the rate of 36 $\text{cm}^{-1} \cdot \text{s}^{-1}$. The signals were detected at room temperature and 760 Torr, when the ADM was filled with 2.5 % water vapor. The SNR value corresponding to the 10 ms, 30 ms and 100 ms were not calculated and plotted in Supplementary Figure 3d, as the LIA detection bandwidth was too narrow to detect the perfect BF-QEPAS signal when the LIA time constant was $>3\text{ms}$.

Supplementary Figure 4: please discuss the different shapes. The figure captions in the main text are much better than in the supplement.

On page 10 in Supplementary, there is a detail discussion about relationship between the wavelength scanning rate of the exciting laser and the BF-QEPAS signal shape:

“When the wavelength is scanned at 18 $\text{cm}^{-1} \cdot \text{s}^{-1}$, the steady state response affects the beat signal and changes the signal shape. When the wavelength scanning rate is too fast, such as 72 $\text{cm}^{-1} \cdot \text{s}^{-1}$, the signal decreased as the energy absorbed by the gas cannot be effectively transformed to acoustic energy. Hence, the optimal wavelength scanning rate was experimentally determined to be 36 $\text{cm}^{-1} \cdot \text{s}^{-1}$ for the detection of water vapor.”

Therefore, we did not add a discussion to the caption of Supplementary Figure 4 to avoid repeating the description.

Supplement line 71: delete one “the”

Done

Supplement figure 7: figure and figure caption are contradictory: in the figure, a scan rate of $50 \text{ cm}^{-1}/\text{s}$ is given, while the caption gives $55 \text{ cm}^{-1}/\text{s}$. Furthermore, line 84 states ICL while line 92 states QCL?

Sorry for the mistakes in the caption of the Supplementary figure 7.

We have replaced the scan rate “ $55 \text{ cm}^{-1}\cdot\text{s}^{-1}$ ” and “QCL” in original manuscript with “ $50 \text{ cm}^{-1}\cdot\text{s}^{-1}$ ” and “DFB-ICL”, respectively.

Line 103: what does “for this purpose” relate to?

We have replaced the sentences on Supplementary page 9, paragraph 1:

“For this purpose, we optimized the LIA filter slope and time constant, respectively.”

with the following one:

“However, the large detection bandwidth results in a high background noise. We optimized LIA filter slope and time constant, respectively, in order to obtain a detection bandwidth which does not distort the BE-QEPAS signal and maintains efficient background noise suppression.”

Beat frequency signal analysis: could you also use the different slopes for a more precise evaluation of the concentration?

That’s a great idea and we tried to use the slopes for evaluating the target gas concentration. However, the relationship between the slope and the gas concentration levels was nonlinear. Therefore, we did not add the corresponding comment in the revised manuscript.

Line 140: add a space before CO.

Done

Line 147/148: sentence is hard to read. Use one sentence for CO and one for CH₄.

We replace the sentence on Supplementary page 11:

“The minimum detection limit and corresponding NNEA coefficient for CO and CH₄ are 220 ppb and $2.37 \times 10^{-9} \text{ cm}^{-1} \cdot \text{W} \cdot \text{Hz}^{-1/2}$, 744 ppm and $1.04 \times 10^{-9} \text{ cm}^{-1} \cdot \text{W} \cdot \text{Hz}^{-1/2}$, respectively.”

with the following one:

“The minimum detection limit and corresponding NNEA coefficient for CO are 10 ppb and $2.3 \times 10^{-8} \text{ cm}^{-1} \cdot \text{W} \cdot \text{Hz}^{-1/2}$, respectively, while they are 40.75 ppm and $1.3 \times 10^{-8} \text{ cm}^{-1} \cdot \text{W} \cdot \text{Hz}^{-1/2}$ for CH₄, respectively.”

Line 171-174: this is a crude method to approximate the derivative – please discuss why it is sufficient/appropriate.

The applicability of any methods depends upon if it is consistent with the experimental results. The method we used obtained a good simulation of the BF-QEPAS signal which is in good agreement with our experimental results. Therefore we believe that this is sufficient for the manuscript.

Since you are using the transient response, there is no standing acoustic wave that could build up within the acoustic resonator. I wonder why you still use it? Please comment on this.

The original reason why we use the acoustic resonator is because we want to compare the performance between the conventional QEPAS and the BF-QEPAS. However we found that the signal of the BF-QEPAS equipped with two micro-tubes is ~9 times larger than that without two micro-tubes. The behavior of signal enhancement from a non-resonant micro-tube is also observed in the conventional QEPAS as described in Ref. 21. The micro-tubes do not exhibit a well-defined resonant behavior. Instead it just confines the acoustic pulse. In the future studies, we will study how the geometrical parameters of the micro-tube affect the BF-QEPAS signal.

References: there are quite a few references cited that hardly touch what is discussed within this paper (other QEPAS-related techniques). Check if you need all of them.

We have deleted the Ref. 7, Ref. 11, and Ref. 25 in the original manuscript. The reference number in the revised manuscript is also changed, accordingly. Furthermore, the Ref. 24 in the original manuscript was cited as Ref. 35 on page 21, paragraph 1 in the revised manuscript: “In this case, the Q-factor of the QTF decreased to 1,850 as the coupling between the QTF and the AmRs^{21,35}.”

Reviewers' comments:

Reviewer #1 (Remarks to the Author):

The authors have satisfactorily responded to all my questions.

Reviewer #2 (Remarks to the Author):

All the comments were addressed and most suggestions implemented into the manuscript. There are two remaining points I am not totally happy with: In reply to the rising part of fig. 3b you state that this is the decaying of the negative valley, which is on first glance plausible. However, then it should have the same time constant as the decaying of the peak since it is governed by the same Q-factor and the same resonant frequency. From the graph I have however the impression that the rising part has a longer time constant than the decaying part and that would not be covered by your explanation.

You answered in your reply letter to the question about the acoustic resonator but added nothing in the manuscript. It would also be interesting for the readers of the paper to know that you will investigate the role of the acoustic resonator further in the future.

line 356/357: sentence sounds weird - omit the "only"

line 522: delete the bar after fig. 1

table 1: NNEA data instead of date

Response to reviewers' comments

Reviewer #1 (Remarks to the Author):

The authors have satisfactorily responded to all my questions.

Reviewer #2 (Remarks to the Author):

All the comments were addressed and most suggestions implemented into the manuscript. There are two remaining points I am not totally happy with: In reply to the rising part of fig. 3b you state that this is the decaying of the negative valley, which is on first glance plausible. However, then it should have the same time constant as the decaying of the peak since it is governed by the same Q-factor and the same resonant frequency. From the graph I have however the impression that the rising part has a longer time constant than the decaying part and that would not be covered by your explanation.

As the 1st harmonic signal has a positive peak and a negative valley, two decaying tails can be observed. However, the first decaying tail from the positive peak is disturbed by the negative valley. It does not reflect a correct response time. The second decaying tail caused by the negative valley is a free exponential decaying and the QTF response time can be obtained from an exponential decay fitting. Therefore, this is why the reviewer has the impression that the decaying tail of the negative valley has a longer time constant than that caused by the positive peak.

To clarify this, we replaced the sentence on page 10:

“Two exponentially decaying tails, related to the peak and the valley...”

with the following one:

“Two decaying tails, related to the peak and the valley...”

We also added the corresponding sentence on page 10:

“But the first decaying tail was disturbed by the valley of the 1st harmonic signal and only the second free decaying tail caused by the valley reflects a correct response time.”

You answered in your reply letter to the question about the acoustic resonator but added nothing in the manuscript. It would also be interesting for the readers of the paper to know that you will investigate the role of the acoustic resonator further in the future.

To clarify this, we replaced the sentence on page 8:

“A fiber-coupled collimator (OZ optics Ltd. Model LPC-01) produced a diode laser beam with a 200 μm diameter, which was directed to an acoustic detection module (ADM) with an

on-beam configuration (see Methods) and avoided touching the QTF³¹”

with the following one:

“A fiber-coupled collimator (OZ optics Ltd. Model LPC-01) produced a diode laser beam with a 200 μm diameter, which was directed to an acoustic detection module (ADM) and avoided touching the QTF³¹.”

We also added the corresponding sentence on page 8:

“For a comparison, the ADM with an on beam configuration (see Methods), which is widely used in conventional QEPAS sensors, is equipped in the BF-QEPAS experimental setup. Although there is no standing acoustic wave built up in the acoustic resonator with the excitation of an acoustic pulse, the behavior of signal enhancement is expected as the non-resonant micro-tube can effectively confine the acoustic pulse²¹.”

We also replaced the sentence on page 21:

“The performance of the BF-QEPAS-based sensor can be further enhanced when it is combined with the overtone resonance mode of a custom fabricated QTF.”

with the following one:

“Further improvement of the detection sensitivity can be achieved either by optimizing the geometrical parameters of the non-resonant micro-tube or by combining with the overtone resonance mode of a custom fabricated QTF.”

line 356/357: sentence sounds weird - omit the "only"

Done

line 522: delete the bar after fig. 1

Done

table 1: NNEA data instead of date

Done

REVIEWERS' COMMENTS:

Reviewer #2 (Remarks to the Author):

The authors have satisfactorily responded to all my questions.

Response to reviewers' comments

Reviewer #2 (Remarks to the Author):

The authors have satisfactorily responded to all my questions.

There is nothing we need to reply.